# Status of Asp29 and Asp40 in the Interaction of *Naja atra* Cardiotoxins with Lipid Bilayers

**DOI:** 10.3390/toxins12040262

**Published:** 2020-04-18

**Authors:** Guan-Lin Wu, Yi-Jun Shi, Chia-Hui Huang, Yuan-Chin Lee, Liang-Jun Wang, Jing-Ting Chiou, Chi-Yu Lu, Long-Sen Chang

**Affiliations:** 1Institute of Biomedical Sciences, National Sun Yat-Sen University, Kaohsiung 804, Taiwan; a10232456z789789@gmail.com (G.-L.W.); d052050012@student.nsysu.edu.tw (Y.-J.S.); nikki781016@hotmail.com (C.-H.H.); d072050008@student.nsysu.edu.tw (Y.-C.L.); d052050011@student.nsysu.edu.tw (L.-J.W.); d042050004@student.nsysu.edu.tw (J.-T.C.); 2Department of Biochemistry, Kaohsiung Medical University, Kaohsiung 807, Taiwan; cylu@kmu.edu.tw; 3Department of Biotechnology, Kaohsiung Medical University, Kaohsiung 807, Taiwan

**Keywords:** cardiotoxin, asp residue, chemical modification, membrane-perturbing activity, membrane-interacted mode

## Abstract

It is widely accepted that snake venom cardiotoxins (CTXs) target the plasma membranes of cells. In the present study, we investigated the role of Asp residues in the interaction of *Naja atra* cardiotoxin 1 (CTX1) and cardiotoxin 3 (CTX3) with phospholipid bilayers using chemical modification. CTX1 contains three Asp residues at positions 29, 40, and 57; CTX3 contains two Asp residues at positions 40 and 57. Compared to Asp29 and Asp40, Asp57 was sparingly modified with semi-carbazide, as revealed by matrix-assisted laser desorption ionization-time of flight (MALDI-TOF) mass and mass/mass analyses. Thus, semi-carbazide-modified CTX1 (SEM-CTX1) mainly contained modified Asp29 and Asp40, while SEM-CTX3 contained modified Asp40. Compared to that of native toxins, trifluoroethanol easily induced structural transition of SEM-CTX1 and SEM-CTX3, suggesting that the structural flexibility of CTXs was constrained by Asp40. Modification of Asp29 and Asp40 markedly promoted the ability of CTX1 to induce permeability of cell membranes and lipid vesicles; CTX3 and SEM-CTX3 showed similar membrane-damaging activity. Modification of Asp residues did not affect the membrane-binding capability of CTXs. Circular dichroism spectra of SEM-CTX3 and CTX3 were similar, while the gross conformation of SEM-CTX1 was distinct from that of CTX1. The interaction of CTX1 with membrane was distinctly changed by Asp modification. Collectively, our data suggest that Asp29 of CTX1 suppresses the optimization of membrane-bound conformation to a fully active state and that the function of Asp40 in the structural constraints of CTX1 and CTX3 is not important for the manifestation of membrane-perturbing activity.

## 1. Introduction

Cardiotoxins (CTXs) are major components of *Elapidae* snake venom and exhibit a variety of pharmacological activities including hemolysis, cardiotoxicity, disruption of muscular integrity, and cytotoxicity [1,2,3,4]. However, the exact mechanisms underlying the broad spectrum of biological activities have not been fully elucidated. Several studies have suggested that CTXs exert their activities through their membrane perforation ability or through the induction of membrane fusion [5]. CTXs are folded into three β-structural loops stabilized by four disulfide bonds. Loops I, II, and III are composed of nonpolar amino acids flanked with positively charged Lys and Arg residues [2]. It has been suggested that the positively charged amino acids of CTXs initiate the interaction with the negatively charged surface of the membrane, subsequently causing the hydrophobic amino acids to form complementary hydrophobic interactions, thus stabilizing the membrane insertion of toxins [5,6].

CTXs are subdivided into S- and P-type CTXs based on their interacted mode with phosphatidylcholine (PC). These CTXs have conserved Ser and Pro residues at positions 28 and 30, respectively [6,7]. P-Type CTXs bind to phospholipid membranes by anchoring two lipid-binding sites in loops I and II, whereas S-type CTXs bind to phospholipid membranes only by their binding site in loop I [7,8]. Computer simulation analyses showed that the loop II region of S-type CTXs is also involved in the formation of the toxin-PC complexes, and that the phospholipid-binding residues of P-type CTXs are different from those of S-type CTXs [6]. Konshina et al. [2] reported that P-type CTXs show higher membrane-damaging activity than that of S-type CTXs on liposomes composed of zwitterionic phospholipid or/and anionic phospholipid vesicles. Sequence alignments showed that *Naja atra* CTX1 and CTX3 belong to the S-type and P-type categories of CTXs, respectively (Figure 1A). CTX1 contains three Asp residues at positions 29, 40, and 57, while CTX3 contains two Asp residues at positions 40 and 57. In CTX1, Asp29 is located at the tip of loop II region; in CTX1 and CTX3, the locations of Asp40 and Asp57 are distant from the tip of loops (Figure 1B). Previous studies have suggested that the salt bridge between Asp57 and Lys2 increases the structural stability of *N. nigricollis* toxin γ, thereby promoting its membrane-perturbing activity [9]. Because the positively charged residues have been implicated in CTX–membrane interactions [2,6], we wanted to explore whether the negatively charged Asp residues inversely modulate the membrane-targeting action of *N. atra* CTX1 and CTX3. To address this question, we investigated the biological and structural properties of Asp-modified CTX1 and CTX3. 

## 2. Results

### 2.1. SEM-CTX1 Contains Modified Asp29 and Asp40

Carboxyl groups in CTX1 and CTX3 were modified with semi-carbazide. Separation of semi-carbazide-modified CTX1 (SEM-CTX1) and semi-carbazide-modified CTX3 (SEM-CTX3) from the reaction mixtures of CTX1 and CTX3 was performed using reverse phase high performance liquid chromatography (HPLC) analyses (Figure 2). In comparison to CTX1 and CTX3, SEM-CTX1 and SEM-CTX3 showed an increase in mass by 109.6 Da and 62.2 Da, respectively (Appendix A). Based to the increase in the molecular weight, two and one semi-carbazide molecules were conjugated with the carboxyl groups of SEM-CTX1 and SEM-CTX3, respectively. Analyses of tryptic hydrolysates of reduced and s-carboxymethylated (RCM)-CTX1 using matrix-assisted laser desorption ionization-time of flight (MALDI-TOF) mass spectrometry showed that the peptides with masses of 1569.1, 948.6, and 1087.6 Da (Figure 3A) corresponded to the fragments at positions 24–36, 37–44 and 51–58, respectively (Table 1). The peptide fragments at positions 24–36, 37–44, and 51–58 contain Asp29, Asp40, and Asp57, respectively (Table 1). The addition of semi-carbazide with Asp resulted in an increase in mass of 57 Da. MALDI-TOF mass analyses showed that three new peptide fragments derived from tryptic-digested RCM-SEM-CTX1 had masses of 1626.6, 1005.6, and 1144.7, respectively (Figure 3A), indicating that the Asp residues in the peptide fragments at positions 24–36, 37–44, and 51–58 were conjugated with semi-carbazide. Notably, in contrast to the peptide (at positions 51–58) with a mass of 1087.6 Da, the peptides with masses of 1569.1 or 948.6 Da were barely detected in the tryptic hydrolysates of RCM-SEM-CTX1 (Figure 3A). It appears that Asp57 is sparingly modified in comparison to Asp29 and Asp40. Chromatographic analysis showed that seven peaks were separated from the tryptic hydrolysates of RCM-CTX1 (Figure 4). MALDI-TOF mass analyses revealed that peaks b, d, and g corresponded to peptide fragments at positions 51–58, 37–44, and 24–36, respectively (Table 1), indicating that peaks b, d, and g contain Asp57, Asp40, and Asp29, respectively. Two new peaks, 1 and 3, were identified from the chromatographic profile of tryptic-digested RCM-SEM-CTX1 (Figure 4). MALDI-TOF mass analyses and NanoUPLC mass/mass analyses indicated that peak 1 corresponded to the peptide at positions 51–58 with semi-carbazide-modified Asp57, while peak 3 corresponded to the peptide at positions 24–36 with semi-carbazide-modified Asp29 (Table 1 and Appendix A). Peak 2 of tryptic-digested RCM-SEM-CTX1 had the same retention time as peak d of tryptic-digested RCM-CTX1 (Figure 4). Peak 2 was identified as the Asp40-modified peptide at positions 37–44 of CTX1, as evidenced by MALDI-TOF mass analyses and mass/mass analyses (Table 1 and Appendix A). Notably, peak b, which contained unmodified Asp57, was still abundantly present in the chromatographic profile of tryptic hydrolysates of RCM-SEM-CTX1 (Figure 4). As SEM-CTX1 contained two semi-carbazide-conjugated Asp residues as revealed by MALDI-TOF mass analyses (Appendix A), these findings allowed us to deduce that SEM-CTX1 mainly represented the derivative with modified Asp29 and Asp40. 

### 2.2. Asp40 Is Identified to Be Modified in SEM-CTX3

The peptide fragments at peaks c and e corresponded to the peptide fragments at positions 51–58 and 37–44 in CTX3 (Figure 4). The amino acid sequences at positions 37–44 and 51–58 of CTX3 were the same as those of CTX1 (Figure 1). MALDI-TOF mass and mass/mass analyses showed that peaks 1 and 2 (derived from the chromatographic profile of tryptic-digested RCM-SEM-CTX3) were the fragments at positions 51–58 and 37–44, respectively. Peaks 1 and 2 contained semi-carbazide-conjugated Asp57 and Asp40, respectively. The Asp40-containing peptide at positions 37–44, with a mass of 948.6 Da, disappeared in the mass spectra of tryptic-digested RCM-SEM-CTX3, while Asp57-containing fragment at positions 51–58 with a mass of 1087.6 Da was still dominantly detected in the mass spectra of tryptic-digested RCM-SEM-CTX3 (Figure 3). Moreover, Asp57-unmodified peptide (peak c) appeared abundantly in the chromatographic profile of the tryptic-digested RCM-SEM-CTX3 (Figure 4). We found that SEM-CTX3 contained one modified Asp residue as revealed by MALDI-TOF analysis (Appendix A). Thus, SEM-CTX3 mainly represented the derivative with semi-carbazide-modified Asp40. 

### 2.3. The Structural Transition of SEM-CTX1 and SEM-CTX3 Response to Trifluoroethanol (TFE)-Induced Effects Are Different from That of Native Toxins

The circular dichroism (CD) spectrum of SEM-CTX1 was distinct from that of CTX1 (Figure 5), suggesting that the secondary structure of CTX1 was altered by the modification of Asp29 and Asp40. However, modification of Asp40 did not appreciably affect the global conformation of CTX3. Previous studies have suggested that trifluoroethanol (TFE)-induced changes in the CD spectra could detect the relative structural stability of CTX isotoxins [9]. Thus, we analyzed the effect of TFE on inducing structural transition of Asp-modified CTX1 and CTX3. TFE is known to increase the helical content of proteins by inducing non-native, partially folded states [10,11]. The representative CD spectra of CTX1 and SEM-CTX1 in the TFE/H_2_O solution consistently showed the transformation of the characters of β-sheet spectra to the characters of the α-helical structure (Figure 6A). An increase in TFE concentration increased the spectral signatures of α-helix with negative bands at 208 nm and 222 nm. The plots of ellipticity as a function of TFE concentration at 222 nm revealed that SEM-CTX1 and SEM-CTX3 showed marked structural transitions at lower TFE concentrations compared to CTX1 and CTX3 (Figure 6B). Because previous studies have shown that the four disulfide bonds of CTXs remain intact during the TFE-induced transition from β-sheet to α-helix [9], our findings suggested that TFE-induced structural transition was related to the structural flexibility of the toxin molecules. Therefore, modification of Asp29 and Asp40 in CTX1 or Asp40 in CTX3 altered their structural characteristics in response to TFE-induced effects. 

### 2.4. Modification of Asp Residues Markedly Increases Membrane-Damaging Activity of CTX1 

Past studies have shown that CTXs damage human red blood cells and cell membranes [4,12]. Since PC, sphingomyelin, and cholesterol (Chol) are the dominant lipid components of mammalian cell membranes [13], lipid vesicles composed of egg-yolk phosphatidylcholine (EYPC), egg-yolk sphingomyelin (EYSM), and/or Chol were created to explore the interaction of CTX1, CTX3, SEM-CTX1, and SEM-CTX3 with membrane bilayers. Representative data showed that CTX3 increased calcein release from EYPC/EYSM/Chol vesicles in a concentration-dependent manner (Figure 7A). Figure 7B illustrates that CTX1 and SEM-CTX1 also increased membrane permeability of EYPC/EYSM vesicles in a concentration-dependent manner. The addition of >300 nM toxins induced a maximal calcein release from the tested lipid vesicles. Treatment with 500 nM CTX1, SEM-CTX1, CTX3, and SEM-CTX3 resulted in the release of calcein entrapped in EYPC/EYSM/Chol vesicles at rates of 2.2 ± 0.7%, 5.7 ± 0.4%, 38.1 ± 0.2%, and 35.3 ± 0.9%, respectively (Inset of Figure 7A). Rates of CTX1-, SEM-CTX1-, CTX3-, and SEM-CTX3-induced leakage of EYPC/EYSM vesicles were 19.8 ± 1.2%, 54.1 ± 2.4%, 70.3 ± 3.4%, and 73.2 ± 2.2%, respectively (Inset of Figure 7B). It appears that incorporation of Chol into lipid vesicles reduced the membrane-damaging activity of CTX1, SEM-CTX1, CTX3, and SEM-CTX3. Previous studies demonstrated that toxin γ, a *N. nigricollis* cardiotoxin, also showed a decrease in membrane-perturbing activity on phospholipid vesicles containing Chol [12]. Notably, modification of Asp40 did not markedly increase the membrane-perturbing activity of CTX3, while blocking the negative charge on Asp29 and Asp40 caused an increase in the membrane-damaging activity of SEM-CTX1. These findings suggested the possibility that Asp29 and Asp40 differently affected membrane-perturbing activity of CTX1. 

### 2.5. Modification of Asp Residues Promotes the Ability of CTX1 to Induce Membrane Permeabilization of K562 Cells

To examine whether modification of Asp residues also promoted the CTX1 cytotoxity on culture cells, we performed calcein assays described in Materials and Methods Section 5.9. The membrane-damaging activities of CTX1 and SEM-CTX1 on parent and Chol-depleted K562 cells loaded with calcein were analyzed here. Chol-depleted K562 cells were prepared by incubation of cells with methyl-β-cyclodextrin (MβCD) as described in Materials and Methods Section 5.9. CTX1 and SEM-CTX1 treatment increased the cell population with a decrease in calcein fluorescent signals (Figure 8A), indicating that CTX1 and SEM-CTX1 had damaged the cell membrane. CTX1 and SEM-CTX1 treatment caused a greater loss of calcein fluorescence in MβCD-treated cells compared to untreated cells (Figure 8B). Furthermore, compared to CTX1, SEM-CTX1 showed a higher capability to induce membrane permeability of either MβCD-treated or untreated cells (Figure 8A,B). These results indicate that modification of Asp residues promotes CTX1-induced membrane permeabilization. 

### 2.6. Native and Carboxyl Group-Modified CTXs Show Similar Lipid-Binding Capability

Studies by Chiou et al. [14] revealed that the interaction of CTXs with *N*-(fluorescein-5-thiocarbamoyl)-1,2-dihexadecanoyl-phosphatidylethanolamine (FPE)-containing lipid vesicles induces changes in FPE fluorescence intensity. The addition of native and modified CTXs consistently increased the FPE fluorescence intensity in FPE-containing vesicles (Figure 9). The binding capability of CTXs for lipid vesicles was calculated from the plot of the FPE fluorescence as a function of CTX concentrations. The dissociation constants (*K*_d_) of CTX1, SEM-CTX1, CTX3, and SEM-CTX3 for EYPC/EYSM/Chol vesicles were 2.6, 1.5, 2.0, and 0.8 μM, respectively. The *K*_d_ of CTX1, SEM-CTX1, CTX3, and SEM-CTX3 for EYPC/EYSM vesicles were 5.9, 4.8, 2.8, and 1.5 μM, respectively. The binding capability of native and Asp-modified CTXs with lipid vesicles could not explain their relative membrane-perturbing potency. Compared to CTX1, SEM-CTX1 notably enhanced the fluorescence signal of FPE/EYPC/EYSM/Chol and FPE/EYPC/EYSM vesicles (Figure 9A,C). Since protein-induced changes in the fluorescence intensity of FPE/lipid vesicles indicate the reorganization events of proteins at the lipid–water interface [15], the present data suggest that SEM-CTX1 and CTX1 adopt different topographical arrangements when they bind with the lipid vesicles. However, SEM-CTX3 and CTX3 differently enhanced the fluorescence intensity of FPE/EYPC/EYSM vesicles but similarly enhanced the fluorescence intensity of FPE/EYPC/EYSM/Chol vesicles (Figure 9B,D). These results suggest that SEM-CTX3 and CTX3 have different membrane-bound modes with EYPC/EYSM vesicles. 

### 2.7. Native and Carboxyl Group-Modified CTXs Adopt Different Conformation for Interacting with Membrane

Some studies have shown that CTXs induce the blue-to-red color transition in lipid/ polydiacetylene (PDA) vesicles, while a greater lipid–surface interaction of CTXs is positively correlated with a higher colorimetric response [12,14]. When compared to CTX1, SEM-CTX1 induced a notable increase in the color transition of the lipid/PDA vesicles (Figure 10A,B), indicating that the geometrical contact of SEM-CTX1 and CTX1 with the tested lipid/PDA vesicles differed. Meanwhile, SEM-CTX3 and CTX3 induced similar changes in the colorimetric response of EYPC/EYSM/Chol and EYPC/EYSM vesicles (Figure 10C,D). These results suggested that the membrane-bound conformation of CTX1 and SEM-CTX1 differed.

## 3. Discussion

Our data revealed that semi-carbazide robustly modifies Asp29 and Asp40 of CTX1 and Asp40 of CTX3 but only marginally modifies Asp57. Previous studies suggested that Asp57 electrostatically interacts with Lys2 in CTXs [9]. The salt bridge may hamper the chemical modification of Asp57. In contrast to CTX1 and CTX3, the structural transition of SEM-CTX1 and SEM-CTX3 notably occurs in the presence of low TFE concentration (Figure 6). Our data highlight that modification of Asp residues reduces the structural constraints of CTX’s response to TFE-induced structural transition. Considering that Asp40 is conserved in CTXs, these results suggest that modification of Asp40 causes CTXs to become more sensitive to the TFE-induced effect. CD measurements revealed that the global structures of SEM-CTX3 and CTX3 are similar, while the gross conformations of SEM-CTX1 and CTX1 differ. This seems to indicate that the integrity of Asp29, rather than Asp40, plays a heavier role in maintaining the native structure of CTX1.

Structural and functional analyses have suggested that the interaction of CTXs with the membrane is mediated through the tips of loops I/II, and that the basic residues flanked by hydrophobic patch at loop I/II regions are involved in the membrane-permeabilizing activity of CTXs [2,4,6,8]. In agreement, previous studies have reported that modification of the conserved Met residues at loop II of CTXs causes a drastic drop in their membrane-damaging activity [16]. Since modification of Asp40 does not profoundly alter the membrane-perturbing activity of CTX3, it is unlikely that the modified Asp40 plays a role in promoting SEM-CTX1-induced membrane leakage. The data from FPE-binding assay and color transition of PDA/lipid suggest that SEM-CTX1 and CTX1 adopt different membrane-bound conformations and topographical arrangements at the lipid-water interface (Figure 9 and Figure 10). Notably, previous studies have suggested that the membrane-bound conformation of CTXs critically modulates their membrane-damaging activity [16,17]. Collectively, these results indicate that the intact Asp29 may functionally modulate the membrane-interacting mode of CTX1, thereby hindering the performance of CTX1 with highly disruptive membrane activity. However, loop I of S-type CTXs has been suggested to be crucial for their binding with zwitterionic phospholipids [4,6]. Gorai et al. [6] proposed that loop II of S-type CTXs is the structural region chiefly governing complex formation of proteins with PC. The finding that modification of Asp29 increases the membrane-damaging activity of CTX1 supports the involvement of loop II in the interaction between S-type CTXs and lipid bilayers. Forouhar et al. [18] suggested that CTXs undergo conformational changes upon binding with membrane and finally oligomerize to form a pore structure, which causes the membrane leakage. Thus, we propose a mechanism where blocking the negative charge at Asp29 renders CTX1 to adopt a more active membrane-interacting conformation. Remarkably, our data suggest that Asp40-controlled structural flexibility in response to the solvation effect of TFE is not important for CTX-induced membrane leakage. Considering that CTXs display a variety of biological activities, the contribution of Asp40 to the other functional actions of CTXs is worth investigating further in future studies. 

## 4. Conclusions

In summary, the data of the present study suggest that Asp29 of CTX1 suppresses the optimization of membrane-bound conformation to a fully active state and that the function of Asp40 in the structural constraints of CTX1 and CTX3 is not important for the manifestation of membrane-perturbing activity.

## 5. Materials and Methods 

### 5.1. Reagents

Purified *N. atra* CTX1 and CTX3 was prepared in our laboratory [19]. Acetonitrile (ACN), calcein, Chol, Discovery BIO wide pore C18 column (4.6 mm × 25 cm), EYPC, EYSM, 1-ethyl-3-(diethylaminopropyl)-carbodiimide hydrochloride (EDC), semi-carbazide, TFA, TFE, 2-(N-morpholino)ethanesulfonic acid (MES) and MβCD were purchased from Sigma-Aldrich Inc. (St. Louis, MO, USA), and sepharose 6B and PD-10 column were from GE Healthcare Life Sciences (Marlborough, MA, USA). Calcein-AM and FPE were obtained from Molecular Probes (Eugene, OR, USA), and 10,12-tricosadiynoic acid was from Alfa Aesar (Ward Hill, MA, USA). Cell culture supplements were the products of GIBCO/Life Technologies Inc. (Carlsbad, CA, USA). 

### 5.2. Preparation of SEM-CTX1 and SEM-CTX3

SEM-CTX1 and SEM-CTX3 were prepared by coupling semi-carbazide to Asp residues of CTX1 and CTX3 through EDC. Briefly, CTX isotoxins (7 mg) were added to a 1 mL stirring solution (10 mM MES, pH 4.0) of semi-carbazide (136 mg), and EDC (12.3 mg/200 μL) was then added. The reaction was allowed to proceed for 3 h at 25 ℃. After desalting on a PD-10 column and lyophilization, the modified proteins were applied on a Discovery BIO wide pore C18 column (4.6 mm × 25 cm), equilibrated with 20% ACN with 0.1% TFA and eluted with a linear gradient to 75% ACN with 0.1% TFA for 50 min. The mass of SEM-CTX1 and SEM-CTX3 was determined using MALDI-7090 MALDI TOF-TOF mass spectrometer (Shimadzu, Japan).

### 5.3. Separation of Tryptic Peptides 

Reduction and S-carboxymethylation of CTXs was conducted according to the procedure described in Chang et al. [20]. The reduced and S-carboxymethylated CTXs in 0.2 M NH_4_HCO_3_ (pH 7.8) were digested by trypsin (30:1, w/w) at 37 ℃for 3 h. The tryptic hydrolysates were applied on a Discovery BIO wide pore C18 column, equilibrated with 5% ACN with 0.1% TFA and eluted with a linear gradient to 50% ACN with 0.1% TFA for 50 min. The peptide fractions were subjected to measure their mass using MALDI TOF-TOF mass spectrometer or analyze their amino acid sequence using NanoUPLC-mass (MS)/MS analyses.

### 5.4. NanoUPLC-MS/MS Analyses

MS/MS analysis was essentially performed according to the procedure described in [21]. In brief, peptide solution was injected into the nanoACQUITY ultra performance liquid chromatography (UPLC) system (Waters, Milford, MA, USA) and detected by linear trap quadropole (LTQ) Orbitrap Discovery hybrid Fourier Transform Mass Spectrometer (Thermo Fisher Scientific Inc., Bremen, Germany) in the positive ion mode at a resolution of 30,000. Mobile phase A (0.1% formic acid) and mobile phase B (100% ACN with 0.1% formic acid) were used to elute the desalting column (Waters Symmetry C18, 5 μm, 180 μm × 20 mm) and analytical column (Waters BEH C18, 1.7 μm, 75 μm × 150 mm). 

### 5.5. Measurement of Circular Dichroism (CD) Spectra

Spectra were recorded on a Jasco model J-810 spectropolarimeter using a cell 0.5 mm path length. CD measurement was performed in 10 mM Tris-HCl-100 mM NaCl (pH 7.5) or TFE/H_2_O solution. The CD spectra were collected from 260–190 nm, and each spectrum was the average of five scans.

### 5.6. Membrane Leakage Induced by CTXs 

EYPC/EYSM/Chol (37/33/30, mol/mol/mol) and EYPC/EYSM (10/9, mol/mol) vesicles encapsulating self-quenching concentration of calcein (50 mM) was prepared essentially in the same manner as described in Yang et al. [22]. The lipid vesicles in 10 mM Tris-HCl-100 mM NaCl (pH 7.5) were used for measuring membrane permeabilizing activity of CTXs. The calcein fluorescence of the liposomes treated with 0.2% Triton X-100 was set for full leakage, and CTX-induced calcein release was expressed as the percentage of full leakage. The calcein fluorescence was excited at 490 nm and the emission was monitored at 520 nm. 

### 5.7. Lipid-Binding Experiments

FPE was incorporated at 2 mol% into EYPC/EYSM or EYPC/EYSM/Chol vesicles according to the procedure of previous studies [22]. FPE fluorescence was recorded with excitation at 490 nm and emission at 515 nm. A plot of 1/ΔF versus 1/CTX gives a straight line where the slope is equal to the dissociation constant of phospholipid–CTX complexes. 

### 5.8. Colorimetric Response of the Phospholipid/PDA Vesicles

Phospholipid/PDA vesicles were prepared from EYPC/EYSM/Chol/10,12-tricosadiynoic acid or EYPC/EYSM/10,12-tricosadiynoic acid in the same manner as described in our previous studies [22]. Vesicle samples at concentration of 0.5 mM (total lipid) in 10 mM Tris-HCl-100 mM NaCl (pH 7.5) were used for experiments. The absorbance at 500 nm (A_500_) and 640 nm (A_640_) of vesicle solution was detected before and after the addition of CTXs. The colorimetric response (%CR) is defined as follows: %CR = ((PB_o_ − PB_1_)/PB_o_ × 100, where PB = A_640_/(A_640_ + A_500_). PB_o_ and PB_1_ are the A_640_/A_500_ ratio before and after addition of CTXs, respectively. 

### 5.9. The Membrane Leakage of Calcein-Labeled Cells

Human leukemia K562 cells were cultured in Roswell Park Memorial Institute (RPMI) 1640 medium supplemented with 10% fetal calf serum, 2 mM glutamine, 100 U/mL penicillin, 100 μg/mL streptomycin, and 1% sodium pyruvate in an incubator humidified with 5% CO_2_. Chol-depleted cells were prepared by incubation of K562 cells (5 × 10^5^ cells/mL) with 2.5 mM MβCD at 37 °C for 1 h. Analyses of cellular Chol content using the Amplex Red Cholesterol Assay kit (Thermo Fisher Scientific, Waltham, MA) showed that the total Chol content of MβCD-treated cells was reduced by approximate 90% in comparison to untreated cells. The calcein-loaded cells were prepared by incubating with 5 μM calcein-AM according to the method described by Chiou et al. [4]. After treatment with CTXs for indicated time periods, the calcein-loaded cells were analyzed using flow cytometric analysis. Reduction in calcein fluorescence represented the release of intracellular calcein from cells.

### 5.10. Statistical Analysis

Statistical analyses were conducted using GraphPad Prism software (La Jolla, CA, USA). All data are reported as mean ± SD. Significant differences among the groups were determined using the unpaired Student’s *t*-test. Difference with *p* value of < 0.05 was considered significant.

## Figures and Tables

**Figure 1 toxins-12-00262-f001:**
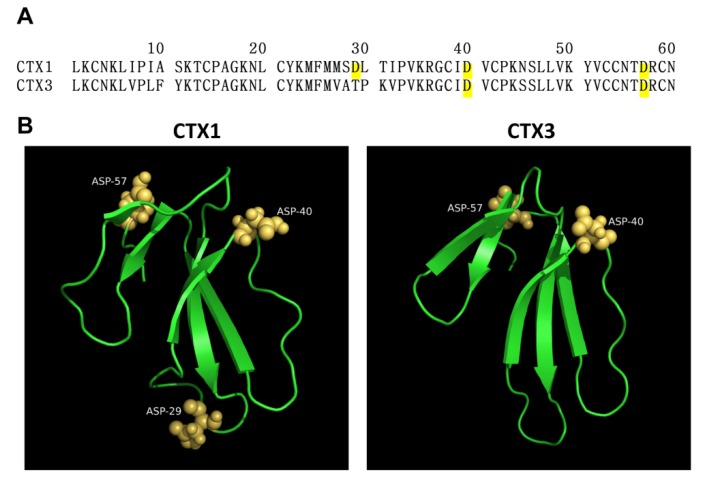
Amino acid sequence and three-dimensional structure of cardiotoxins (CTXs) CTX1 and CTX3. (**A**) Sequence alignments of CTX1 and CTX3. (**B**) Three-dimensional structure of CTX1 (PDB ID 2CDX) and CTX3 (PDB ID 2CRT) showing the spatial positions of Asp29, Asp40, and Asp57.

**Figure 2 toxins-12-00262-f002:**
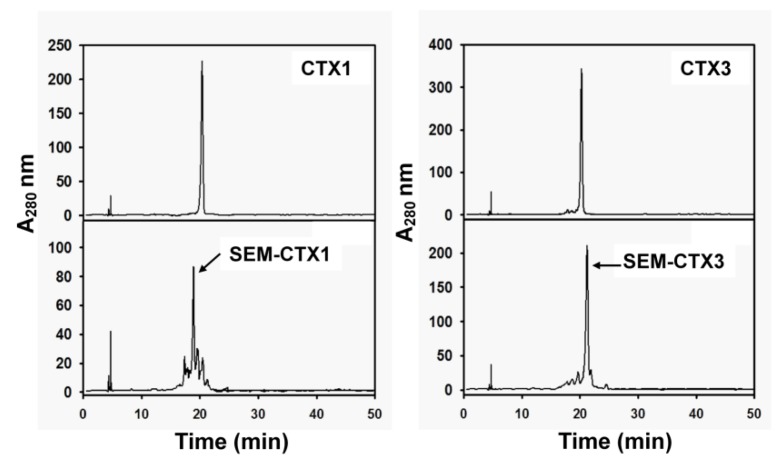
Separation of semi-carbazide-modified (SEM)-CTX1 and SEM-CTX3 on a Discovery BIO wide pore C18 column. The column (4.6 mm × 25 cm) equilibrated with 0.1% trifluoroacetic acid (TFA) was eluted with a linear gradient of 20–75% acetonitrile for 50 min. The flow rate was 0.8 mL/min, and the effluent was monitored at 280 nm.

**Figure 3 toxins-12-00262-f003:**
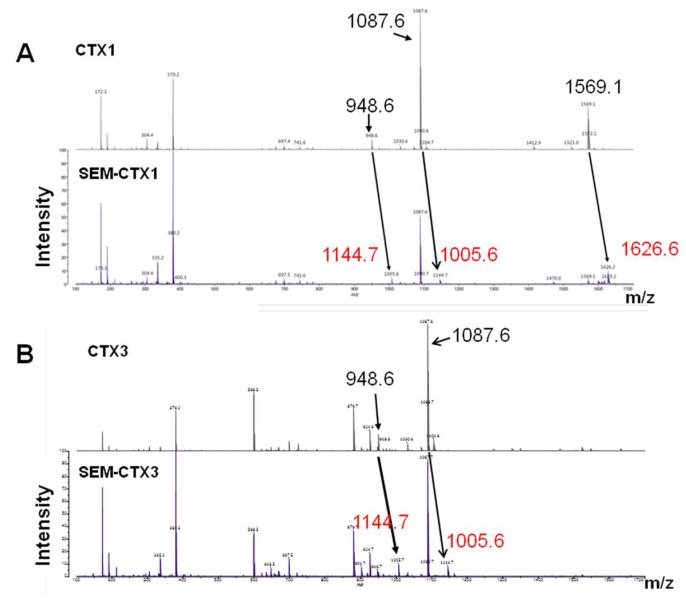
Matrix-assisted laser desorption ionization-time of flight (MALDI-TOF) mass analyses of tryptic digested CTX1 and tryptic digested CTX3. The addition of semi-carbazide caused an increment of 57 Da. The mass of peptides with modified Asp in (**A**) tryptic-digested CTX1 and (**B**) tryptic-digested CTX3 were indicated by red letters.

**Figure 4 toxins-12-00262-f004:**
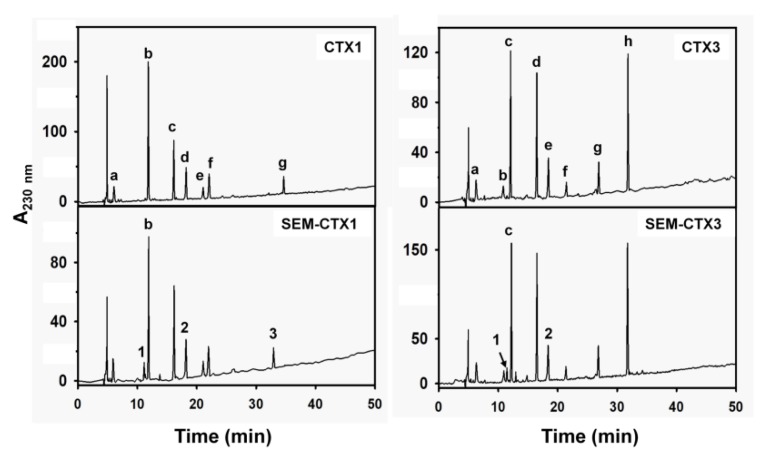
Separation of tryptic peptides from native and semi-carbazide-modified CTXs. Tryptic hydrolysates of reduced and s-carboxymethylated (RCM)-CTX1, RCM-SEM-CTX1, RCM-CTX3, and RCM-SEM-CTX3 were applied on a Discovery BIO wide pore C18 column (4.6 mm × 25 cm) equilibrated with 0.1% TFA was eluted with a linear gradient of 5–50% acetonitrile for 50 min. The flow rate was 0.8 mL/min, and the effluent was monitored at 230 nm.

**Figure 5 toxins-12-00262-f005:**
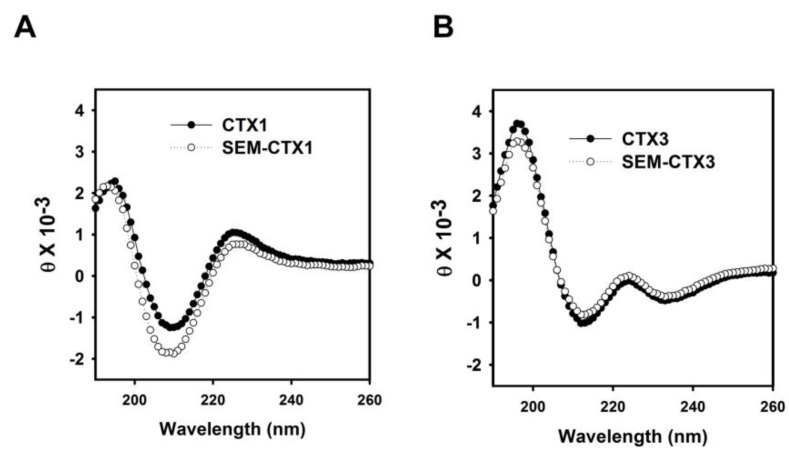
Circular dichroism (CD) spectra of (**A**) CTX1, SEM-CTX1 and (**B**) CTX3, SEM-CTX3.

**Figure 6 toxins-12-00262-f006:**
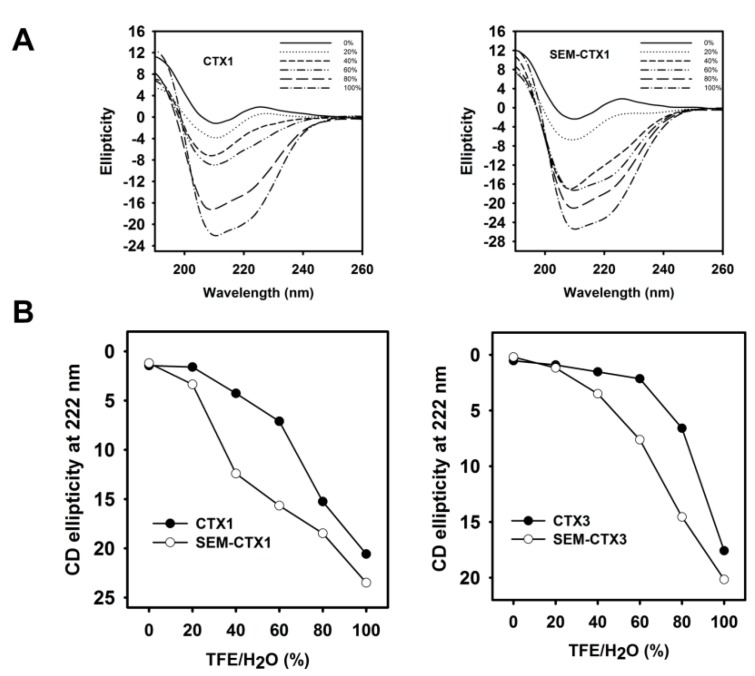
Trifluoroethanol (TFE)-induced structural transition of CTX1, SEM-CTX1, CTX3, and SEM-CTX3. (**A**) TFE induced an increase in the negative ellipticity of native and modified CTX1. (**B**) The plots showed an increased in the negative ellipticity at 222 nm with increasing TFE concentration.

**Figure 7 toxins-12-00262-f007:**
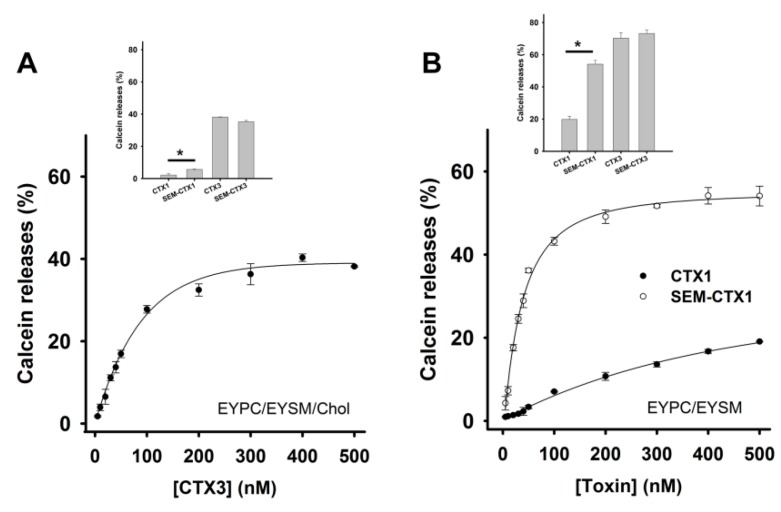
Effect of CTX1, SEM-CTX1, CTX3, and SEM-CTX3 on the permeability of egg-yolk phosphatidylcholine/egg-yolk sphingomyelin (EYPC/EYSM) and EYPC/EYSM/cholesterol (Chol) vesicles. The signal was expressed as the percentage of total calcein release after addition of 0.2% Triton X-100. (**A**) CTX3 concentration-dependently induced the release of calcein from EYPC/EYSM/Chol vesicles. (Inset) Membrane-damaging activity of 500 nM CTX1, SEM-CTX1, CTX3, and SEM-CTX3 on EYPC/EYSM/Chol (* *p* < 0.05). (**B**) CTX1 and SEM-CTX1 concentration-dependently induced the release of calcein from EYPC/EYSM vesicles. (Inset) Membrane-damaging activity of 500 nM CTX1, SEM-CTX1, CTX3, and SEM-CTX3 on EYPC/EYSM vesicles (* *p* < 0.05).

**Figure 8 toxins-12-00262-f008:**
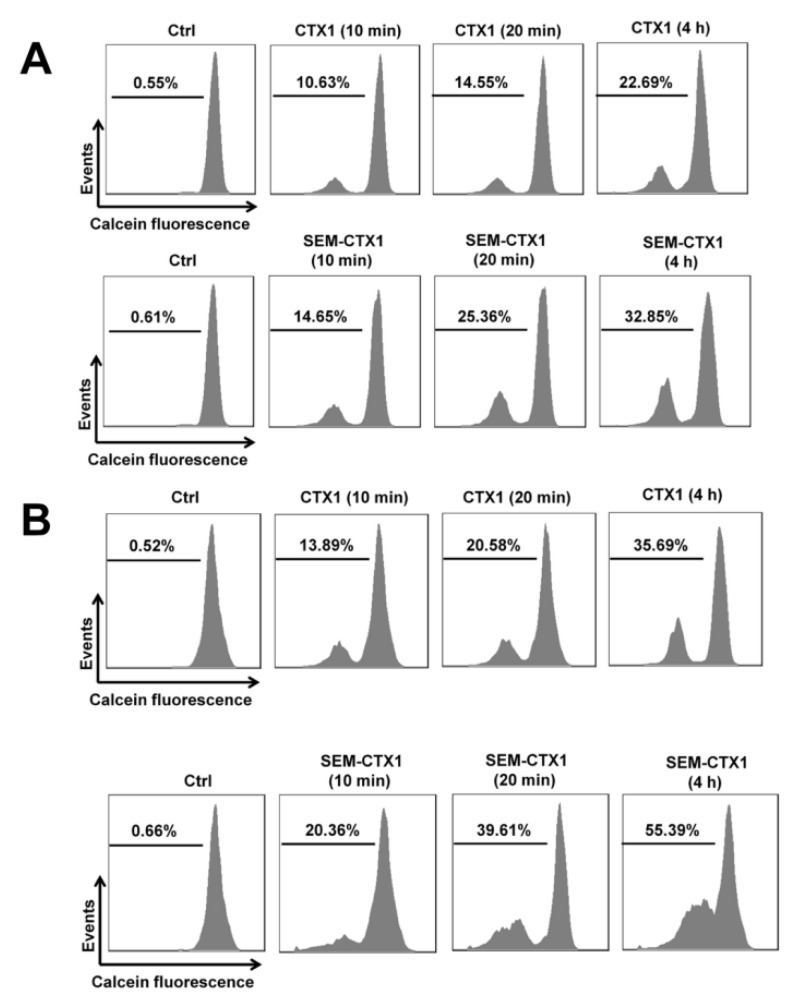
CTX1 and SEM-CTX1 induced membrane permeability of methyl-β-cyclodextrin (MβCD)-treated and -untreated K562 cells. (**A**) K562 cells and (**B**) MβCD-treated K562 cells were incubated with calcein acetoxymethyl ester (calcein-AM) according to the procedure described in Materials and Methods section. The calcein-loaded cells were treated with 500 nM CTXs for indicated time periods, and then were analyzed by flow cytometry. The cell population with reduced fluorescence intensity represented the release of intracellular calcein.

**Figure 9 toxins-12-00262-f009:**
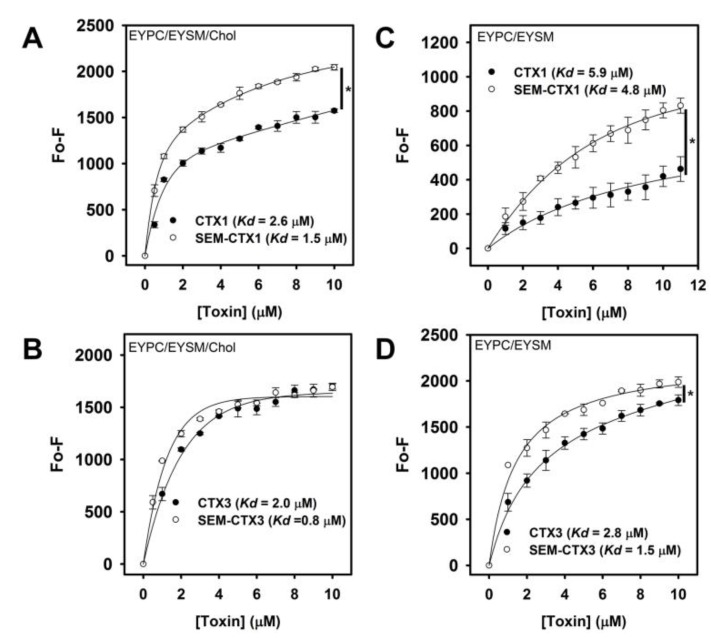
The binding capability of CTX1, SEM-CTX1, CTX3, and SEM-CTX3 with phospholipid vesicles. Binding of CTX1, SEM-CTX1, CTX3, and SEM-CTX3 with phospholipid vesicles enhanced the fluorescence intensity of and (**A**,**B**) N-(fluorescein-5-thiocarbamoyl)-1,2-dihexadecanoyl- phosphatidylethanolamine (FPE)/EYPC/EYSM/Chol and (**C**,**D**) FPE/EYPC/EYSM vesicles (mean ± standard deviation (SD), * *p* < 0.05). The used lipid concentration was 2.5 μM. Fluorescence emission intensity at 510 nm was measured.

**Figure 10 toxins-12-00262-f010:**
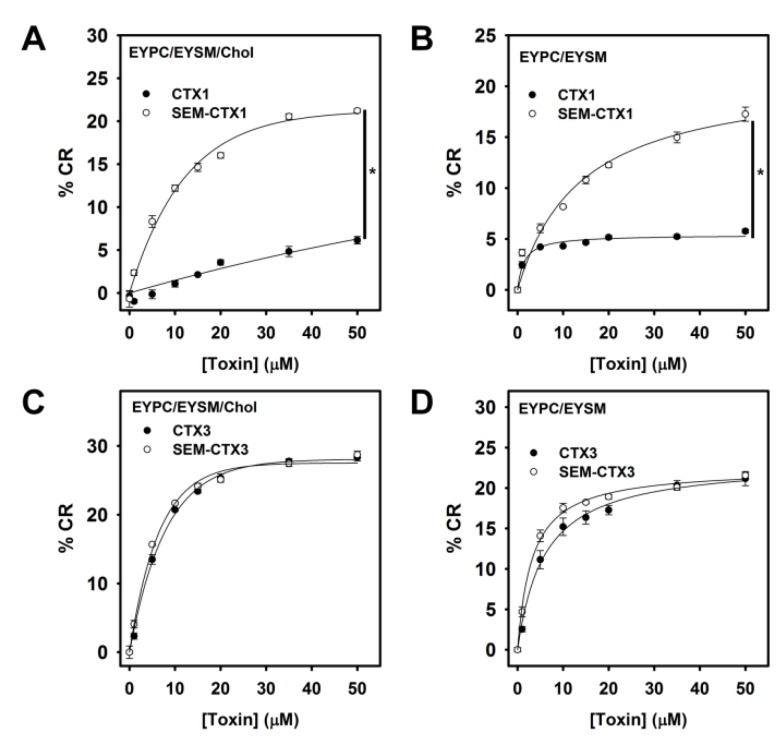
Colorimetric dose-response curve of PDA/EYPC/EYSM and PDA/EYPC/EYSM/Chol vesicles titrated with CTX1, SEM-CTX1, CTX3, and SEM-CTX3. The experiments were conducted essentially according to the procedure described in Materials and Methods section. Each value is the mean ± SD of three independent experiments (* *p* < 0.05). The total lipid concentration of (**A**,**C**) PDA/EYPC/EYSM/Chol and (**B**,**D**) PDA/EYPC/EYSM solution was 0.5 mM.

**Table 1 toxins-12-00262-t001:** The mass of semi-carbazide-conjugated peptide fragments in semi-carbazide-modified cardiotoxin 1 (SEM-CTX1) and SEM-CTX3.

Tryptic Peptides	Positions ^(#)^	Predicted Mass (Da)	Measured Mass (Da)
**CTX1**			
MFMMSDLTIPVKR	24–36 (Asp29)	1569	1569.1
GC^(a)^IDVC^(a)^PK	37–44 (Asp40)	950	948.6
YVC^(a)^C^(a)^NTDR	51–58 (Asp57)	1089	1087.6
**SEM-CTX1**			
MFMMSD^(b)^LTIPVKR	24–36 (Asp29)	1626	1626.6
GC^(a)^ID^(b)^VC^(a)^PK	37–44 (Asp40)	1007	1005.6
YVC^(a)^C^(a)^NTD^(b)^R	51–58 (Asp57)	1146	1144.7
**CTX3**			
GC^(a)^IDVC^(a)^PK	37–44 (Asp40)	950	948.6
YVC^(a)^C^(a)^NTDR	51–58 (Asp57)	1089	1087.6
**SEM-CTX3**			
GC^(a)^ID^(b)^VC^(a)^PK	37–44 (Asp40)	1007	1005.6
YVC^(a)^C^(a)^NTD^(b)^R	51–58 (Asp57)	1146	1144.7

^(a)^ The sulfhydryl group of the cysteine residue is conjugated with -CH_2_-CONH_2_. ^(b)^ The carboxyl group of the aspartate residue is conjugated with -NH-NH-CONH_2_. ^(#)^ The numbers indicate the positions of peptide fragments in the amino acid sequence of CTX1 and CTX3, and the parentheses indicate the aspartate residues in the peptide fragments.

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
