# Peer review of "Status of Asp29 and Asp40 in the Interaction of Naja atra Cardiotoxins with Lipid Bilayers"

_toxins, 2020, doi:10.3390/toxins12040262_

Round 1

Reviewer 1 Report

There are two parts in the manuscript. The 1st part mainly described the Asp modification, authentication, and their physical characteristics by using MALDI-TOF mass, HPLC, CD and so on. In the 2nd part, authors examined the role of CTXs in K-562 cells and also examined the mechanism how CTXs bind the lipid bilayer.

I have several comments to authors:

  1. As author mentioned, I understand that addition of semicarbazide with Asp resulted in an increase in mass of 57 Da since OH leaving from carboxyl group and H leaving from NH2 gave a mass of 57 Da (75- 18 = 57). I also understand the mass of 1569, 948, 1087 shown in Figure 3A should come from 1569, 948, and 1087, respectively. However, what are 109.6 Da and 62.2 Da described in the first paragraph in the Results section?

  1. It is quite abruptly about the calcein assays. It would be better if you can give introduction to readers who are unfamiliar with the assays. For example, to investigate the CTX cytotoxicity, we performed calcein assays described in the Materials and Methods section 5.9 ... as you like.

  1. What is the rational of using K-562 bone marrow cells for the validation? CTX is more sensitive to blood cells or other types of cells, I presume, as you mentioned in the manuscript.

  1. It is difficult for me to buy your conclusion that "Notably, 
modification of Asp40 did not markedly increase the membrane-perturbing activity of CTX3. Thus, it was inferred that blocking the negative charge on Asp29 (rather than on Asp40) caused an increase
in the membrane-damaging activity of SEM-CTX1." If you would like to conclude as abovementioned, you need to use CTX1 mutations that one has Asp29Ala and one has Asp40Ala, and then, you can compare the result obtained from theses mutations with the side-by-side experiment as a rigorous experimental design. Comparing CTX1 with CTX3 is misleading.

  1. There are a few typo, please double-check your manuscript.

Author Response

Comment 1: As author mentioned, I understand that addition of semicarbazide with Asp resulted in an increase in mass of 57 Da since OH leaving from carboxyl group and H leaving from NH2 gave a mass of 57 Da (75- 18 = 57). I also understand the mass of 1569, 948, 1087 shown in Figure 3A should come from 1569, 948, and 1087, respectively. However, what are 109.6 Da and 62.2 Da described in the first paragraph in the Results section?

We agree reviewer’s opinion.  The conjugation of one and two semicarbazide with carboxyl groups of CTXs should increase the mass of 57 Da and 114 Da, respectively.  However, mass analyses showed that, in comparison to CTX1 and CTX3, SEM-CTX1 and SEM-CTX3 showed an increase in mass by 109.6 Da and 62.2 Da, respectively.  Since the measurement of m/z can be affected by experimental parameters that include e.g., calibration coefficients, ion intensity, and temperature changes during the measurement [1].  We speculate that the deviation between measured mass and predicted mass arises from experimental conditions of mass measurement.  Since the molecule masses of these proteins were measured from our experimental analyses, we truly presented these data in our manuscript.  Nevertheless, the results of mass analyses allows us to deduce that one and two semicarbazide molecules conjugated with carboxyl groups of CTX1 and CTX3, respectively.

[1] Petyuk VA, Jaitly N, Moore RJ, Ding J, Metz TO, Tang K, Monroe ME, Tolmachev AV, Adkins JN, Belov ME, Dabney AR, Qian WJ, Camp DG 2nd, Smith RD. Elimination of systematic mass measurement errors in liquid chromatography-mass spectrometry based proteomics using regression models and a priori partial knowledge of the sample content. Anal Chem 2008; 80:693-706.

Comment 2: It is quite abruptly about the calcein assays. It would be better if you can give introduction to readers who are unfamiliar with the assays. For example, to investigate the CTX cytotoxicity, we performed calcein assays described in the Materials and Methods section 5.9 ... as you like.

The sentences on lines 217-221 have been revised according to reviewer’s suggestion.  

Comment 3: What is the rational of using K-562 bone marrow cells for the validation? CTX is more sensitive to blood cells or other types of cells, I presume, as you mentioned in the manuscript.

Our previous studies have shown that membrane-active protein induces higher membrane permeability on calcein-loaded K562 cells than on calcein-loaded breast cancer MCF-7 cells [1].  Furthermore, some studies have shown that K562 cells are sensitive to the cardiotoxin cytotoxicity [2,3].  Thus, we analyze the capability of native and modified CTX1 to induce membrane permeability of K562 cells. 

[1]Yang SY, Chen YJ, Kao PH, Chang LS. Bovine serum albumin with glycated carboxyl groups shows membrane-perturbing activities. Arch Biochem Biophys 2014; 564:43-51.

[2] Yang SH, Chien CM, Lu MC, Lu YJ, Wu ZZ, Lin SR. Cardiotoxin III induces apoptosis in K562 cells through a mitochondrial-mediated pathway. Clin Exp Pharmacol Physiol 2005; 32:515-520.

[3] Yang SH, Lu MC, Chien CM, Tsai CH, Lu YJ, Hour TC, Lin SR. Induction of apoptosis in human leukemia K562 cells by cardiotoxin III. Life Sci 2005; 76:2513-2522.

Comment 4: It is difficult for me to buy your conclusion that "Notably, 
modification of Asp40 did not markedly increase the membrane-perturbing activity of CTX3. Thus, it was inferred that blocking the negative charge on Asp29 (rather than on Asp40) caused an increase
in the membrane-damaging activity of SEM-CTX1." If you would like to conclude as abovementioned, you need to use CTX1 mutations that one has Asp29Ala and one has Asp40Ala, and then, you can compare the result obtained from theses mutations with the side-by-side experiment as a rigorous experimental design. Comparing CTX1 with CTX3 is misleading.

Thank you very much for reviewer’s valuable suggestion.  The sentences (lines 202-204) have been changed as “Notably, modification of Asp40 did not markedly increase the membrane-perturbing activity of CTX3, while blocking the negative charge on Asp29 and Asp40 caused an increase in the membrane-damaging activity of SEM-CTX1.  These findings suggested the possibility that Asp29 and Asp40 differently affected membrane-perturbing activity of CTX1. “. 

Comment 5:There are a few typo, please double-check your manuscript.

The typo has been carefully amended.

Reviewer 2 Report

The authors present a thorough exploration of the functional significance of the Asp residue in snake venom cardiotoxins. The manuscript is well-written and provides a significant contribution to our understanding of cardiotoxin biochemistry. I have only a few minor suggestions below:

Because the results are very data/experiment heavy, I suggest adding sub-headings with brief descriptions of the main finding for each section to give readers a heads-up about what to expect in each sub-section.

L55 – change “want” to “wanted”

L83 – change “thta” to “that”

L233 – change sentence from past to present tense

Figs 9 and 10 – add Kd values and indicate statistically significant differences in the graphs

L277 – change “provide insight into the fact that” to “indicate that”

L369 – in what software/package were data analyzed?

Author Response

Comment 1: Because the results are very data/experiment heavy, I suggest adding sub-headings with brief descriptions of the main finding for each section to give readers a heads-up about what to expect in each sub-section.

The subtitles have been included in Results section

Comment 2: L55 – change “want” to “wanted”

The mistake (line 58) has been revised according to reviewer’s syggestion.

Comment 3 : L83 – change “thta” to “that”

The mistake (line 88) has been revised according to reviewer’s syggestion.

Comment 4:L233 – change sentence from past to present tense

The sentence (lines 266-267) has been revised according to reviewer’s syggestion.

Comment 5:Figs 9 and 10 – add Kd values and indicate statistically significant differences in the graphs

Fig. 9 and Fig. 10 have been revised according to reviewer’s suggestion.

Comment 6: L277 – change “provide insight into the fact that” to “indicate that”

The sentence (line 313) has been revised according to reviewer’s syggestion.

Comment 7: L369 – in what software/package were data analyzed?

Statistical analyses were conducted using GraphPad Prism software (La Jolla, CA, USA).  A new sentence (line 417) has been inserted in revised manuscript.